# Maternal Uniparental Isodisomy of Chromosome 4 and 8 in Patients with Retinal Dystrophy: *SRD5A3*-Congenital Disorders of Glycosylation and *RP1*-Related Retinitis Pigmentosa

**DOI:** 10.3390/genes13020359

**Published:** 2022-02-16

**Authors:** Nobutaka Tachibana, Katsuhiro Hosono, Shuhei Nomura, Shinji Arai, Kaoruko Torii, Kentaro Kurata, Miho Sato, Shuichi Shimakawa, Noriyuki Azuma, Tsutomu Ogata, Yoshinao Wada, Nobuhiko Okamoto, Hirotomo Saitsu, Sachiko Nishina, Yoshihiro Hotta

**Affiliations:** 1Department of Ophthalmology, Hamamatsu University School of Medicine, Hamamatsu 431-3192, Japan; reenobu@hama-med.ac.jp (N.T.); hosono@hama-med.ac.jp (K.H.); 41246273@hama-med.ac.jp (S.N.); shinji@hama-med.ac.jp (S.A.); kaorukot@hama-med.ac.jp (K.T.); kurata@hama-med.ac.jp (K.K.); mihosato@hama-med.ac.jp (M.S.); 2Department of Pediatrics, Osaka Medical and Pharmaceutical University Hospital, Takatsuki 569-8686, Japan; shuuichi.shimakawa@ompu.ac.jp; 3National Center for Child Health and Development, Department of Ophthalmology and Laboratory for Visual Science, Tokyo 157-8535, Japan; azuma-n@ncchd.go.jp (N.A.); nishina-s@ncchd.go.jp (S.N.); 4Department of Biochemistry, Hamamatsu University School of Medicine, Hamamatsu 431-3192, Japan; tomogata@hama-med.ac.jp (T.O.); hsaitsu@hama-med.ac.jp (H.S.); 5Hamamatsu Medical Center, Department of Pediatrics, Hamamatsu 432-8580, Japan; 6Department of Molecular Medicine, Osaka Women’s and Children’s Hospital, Izumi 594-1101, Japan; waday@wch.opho.jp (Y.W.); genetics@wch.opho.jp (N.O.); 7Department of Medical Genetics, Osaka Women’s and Children’s Hospital, Izumi 594-1101, Japan

**Keywords:** uniparental isodisomy, retinal dystrophy, congenital disorders of glycosylation, *SRD5A3* gene, RP1-related retinitis pigmentosa

## Abstract

Purpose: Uniparental disomy (UPD) is a rare chromosomal abnormality. We performed whole-exosome sequencing (WES) in cases of early-onset retinal dystrophy and identified two cases likely caused by UPD. Herein, we report these two cases and attempt to clarify the clinical picture of retinal dystrophies caused by UPD. Methods: WES analysis was performed for two patients and their parents, who were not consanguineous. Functional analysis was performed in cases suspected of congenital disorders of glycosylation (CDG). We obtained clinical case data and reviewed the literature. Results: In case 1, a novel c.57G>C, p.(Trp19Cys) variant in *SRD5A3* was detected homozygously. Genetic analysis suggested a maternal UPD on chromosome 4, and functional analysis confirmed CDG. Clinical findings showed early-onset retinal dystrophy, intellectual disability, and epilepsy. In case 2, an Alu insertion (c.4052_4053ins328, p.[Tyr1352Alafs]) in *RP1* was detected homozygously. Maternal UPD on chromosome 8 was suspected. The clinical picture was consistent with *RP1*-related retinitis pigmentosa. Although the clinical features of retinal dystrophy by UPD may vary, most cases present with childhood onset. Conclusions: There have been limited reports of retinal dystrophy caused by UPD, suggesting that it is rare. Genetic counseling may be encouraged in pediatric cases of retinal dystrophy.

## 1. Introduction

Uniparental disomy (UPD) is a rare event in which an individual with a diploid genome carries either two homologs of a pair of chromosomes from one parent (uniparental heterodisomy), or two copies of a single chromosome from one parent (uniparental isodisomy) [1]. It is divided into paternal and maternal UPD, depending on whether the homologous chromosomes are derived from the father or mother, respectively [2]. In 1988, Spence et al. [3] reported the first UPD (chromosome 7) case presenting with cystic fibrosis and short stature. The major mechanism of UPD formation is trisomic rescue [4]. Approximately 100 genes around specific parts of chromosomes 6, 7, 11, 14, 15, and 20 have lost their functions due to imprinting; therefore, the UPD of the unfunctional allele causes congenital disorders (Prader–Willi syndrome and Kagami–Ogata syndrome, among others) [5,6,7]. In the case of a recessive genetic disease, even if only one parent is a carrier, the disease may develop because of the overlapping variants present in the child. Engle et al. [8] reported that UPD causes 40 recessive genetic diseases on 13 autosomal and XY chromosomes. There have been few reports of retinal dystrophy due to UPD (including whole and partial UPD) on chromosomes 1, 2, 6, and 14 [9,10,11,12,13,14,15,16,17]. In this study, we present two cases of early-onset retinal dystrophy, including congenital disorders of glycosylation (CDG), caused by maternal UPD, examined via whole-exome sequencing (WES). We report detailed clinical features of the two cases and review previous reports of retinal dystrophy cases with UPD, obtained via a search of PubMed, to characterize their clinical features.

## 2. Materials and Methods

### 2.1. Ethics Statement

This study was approved by the Institutional Review Board for Human Genetic and Genome Research at the Hamamatsu University School of Medicine (permit no. 14-040). All study procedures adhered to the tenets of the Declaration of Helsinki. Written informed consent was obtained from the patients and their parents after all the study procedures had been explained in detail.

### 2.2. Genetic Analysis

Genomic DNA was captured using a SureSelect Human All ExonV6 kit (Agilent Technologies, Santa Clara, CA, USA) and sequenced on a NextSeq500 (Illumina, San Diego, CA, USA) system with 151 bp paired-end reads. Sequenced reads were aligned to the GRCh38 reference genome using BWA-mem (Version 0.7.17) with default parameters. The removal of duplicated reads and base quality recalibration were performed with GATK Version 4.1.9.0 using MarkDuplicatesSpark, BaseRecalibrator, and ApplyBQSR. Variants were identified using HaplotypeCaller, and raw variants were filtered out when their parameters met any of the following values: QD < 2.0, MQ < 30.0, FS > 60.0, MQRankSum < −12.5, or ReadPosRankSum < −8.0 for single nucleotide variants; and QD < 2.0, ReadPosRankSum < −20.0, or FS > 200.0 for insertions/deletions. Final variants were annotated using Annovar [18] to estimate the predictive value of the functional impact of the coding variants and assess the allele frequency: an in-house database of 82 control exomes, 8.3KJPN (https://jmorp.megabank.tohoku.ac.jp/, accessed on 3 September 2021) [19], and gnomAD v3.1.1 database (https://gnomad.broadinstitute.org/, accessed on 3 September 2021) were used [20]. Splicing junction prediction was performed using SpliceAI [21]. We defined “rare candidate variants” as a minor allele frequency equal to or less than 0.01 in the three databases and a quality score greater than 200 points. Synonymous and splicing variants within 10 bp from the exon–intron border were retained as candidates if the SpliceAI score was greater than or equal to 0.2. Loss of heterozygosity (LOH) was detected using H3M2 by calculating the B allele frequency (BAF) using WES data [22]. The LOH region was determined by the absence of heterozygous alleles, representing a BAF value of approximately 0.5. De novo assembly of the clipped reads was performed using Megahit v1.2.9 [23].

The Alu element insertion could not be efficiently detected by the exon capture process of WES. Therefore, after WES, we performed polymerase chain reaction (PCR)-based screening for Alu element insertion, in which no pathogenic variant was identified by WES. The following primer set was used for the detection of Alu element insertion: exon 4 forward primer 5′-TGTGCTCAAAAGGA-GAACCATAC- 3′ and reverse primer 5′-TCCTGAAACTTCCTTAGTGAAC-3′. The Alu element insertion was confirmed by size differences (expected sizes of the Alu element insertion and wild type were 675 bp and 347 bp, respectively) in PCR products on electrophoresis.

### 2.3. Electrospray Ionization Mass Spectrometry

Electrospray ionization mass spectrometry (ESI-MS) of transferrin immunopurified from serum was performed according to a previously reported method [24]. Briefly, an affinity column was prepared using a rabbit polyclonal antibody against human transferrin (DAKO, Grostrup, Denmark) and a ligand-coupled Sepharose column (HiTrap NHS-activated HP; GE Healthcare, Piscataway, NJ, USA); then, the antibody-coupled Sepharose was recovered from the column. Subsequently, 10 μL of serum was mixed with a 20 μL slurry of the antibody-coupled Sepharose in 0.5 mL of phosphate-buffered saline (PBS), and the solution was incubated at 4 °C for 30 min. After washing in PBS, transferrin was eluted from Sepharose in 0.1 M glycine–HCl buffer at pH 2.5. For liquid chromatography–mass spectrometry, the immunopurified transferrin was injected into a C4 reversed-phase column (2 mm diameter and 10 mm length) connected to an API4500 quadrupole mass spectrometer (Sciex, Framingham, MA, USA) and eluted with a 60% acetonitrile/0.1% formic acid solution. The ESI mass spectrums of the multiply charged ions were transformed into a single-charge spectrum using Promass protein deconvolution software (ThermoFisher Scientific, Waltham, MA, USA).

### 2.4. Clinical Assessment

We performed comprehensive ophthalmic examinations, including decimal best-corrected visual acuity (BCVA) measurement, Goldmann perimetry, fundus photography, fundus autofluorescence imaging (FAF) using a Spectralis HRA (Heidelberg Engineering, Heidelberg, Germany), and optical coherence tomography (OCT; Heidelberg Engineering). Full-field electroretinography (ERG) was recorded in accordance with the protocols of the International Society for Clinical Electrophysiology of Vision [25]. In addition, we reviewed and evaluated previous studies describing retinal dystrophy caused by UPD.

## 3. Results

### 3.1. Case 1

#### 3.1.1. Clinical Characteristics

Case 1 was a 14-year-old girl with non-consanguineous parents. The main complaints were nystagmus and visual impairment. Nystagmus and motor developmental regression were observed at birth. The patient was examined by an ophthalmologist who noted a refractive error and prescribed eyeglasses. The patient was admitted to our hospital at the age of 8 years. She had an intellectual disability and had been prophylactically treated for epilepsy since the age of 2 years. There were no special notes on her family history. Her BCVA at the age of 8 years was 0.1 in each eye, which then deteriorated to 0.08 by the age of 14 years. The visual field observed at the age of 14 years was remarkably constricted in comparison with that observed at the age of 9 years (Figure 1A–D). Fundus examination showed poor retinal color and narrowing of the retinal blood vessels (Figure 1E,F). Vitreous degeneration was observed at 14 years of age. Severe myopia and astigmatism (−8.00D cyl −3.50D) were complicated with almost the same acuity from the ages of 8 to 14 years. Although OCT images were poor due to nystagmus, retinal thinning was observed, and the ellipsoid zone (EZ) line disappeared except for the fovea (Figure 1G,H). Both the rod and cone responses of ERG disappeared from early childhood. ERG at 9 years of age showed an extinguishing pattern (Appendix A).

#### 3.1.2. Genetic Studies

Trio (patient and her parents)-based WES of family 1 revealed LOH stretches on the whole of chromosome 4, where heterozygous alleles representing a BAF value of approximately 0.5 were absent (Figure 2A). We found 10 rare homozygous candidate variants on chromosome 4, and parental genotypes of these variants indicated maternal uniparental isodisomy (Figure 2A). Four of the ten genes have been associated with human disorders (Appendix A). Among them, *SRD5A3* was of interest because congenital eye malformations, including variable visual loss, have been reported in patients with homozygous *SRD5A3* variants [26]. *SRD5A3* encodes steroid 5α-reductase 3, which is essential for N-glycosylation by converting polyprenol to dolichol [26]. The *SRD5A3* variant (NM_024592.5:c.57G>C, p.[Trp19Cys]) substituted a tryptophan residue, which is conserved among vertebrates and located in the transmembrane domain (Figure 2B), and was predicted to be damaging by in silico prediction tools (Appendix A).

#### 3.1.3. Mass Spectrum of Transferrin

The *SRD5A3* is necessary for the reduction in polyprenol, which is the major pathway for dolichol biosynthesis. N-glycans are formed on dolichol before being transferred to the recipient proteins; therefore, the metabolic block leads to defects in the early N-glycosylation pathway. To examine the mutational effect on N-glycosylation, serum transferrin was analyzed by ESI-MS. In the deconvoluted spectrum of transferrin from the patient, molecules with a single glycan and without glycans were detected, indicating type-1 CDG due to the metabolic block occurring early in the N-glycosylation pathway (Figure 2C).

### 3.2. Case 2

#### 3.2.1. Clinical Characteristics

Case 2 was a 31-year-old man with non-consanguineous parents. The main complaints were photophobia and night blindness experienced at the age of 7 years. He was diagnosed with RP at the age of 10 years. He had been followed-up at our hospital since the age of 26 years. He had no systemic complications or relevant family history. His BCVA at the age of 26 years was 0.07 in the right eye and 0.2 in the left eye, with moderate myopia and astigmatism (−5.00D cyl −2.00D). The visual field was highly constricted (Figure 3A,B). A fundus examination revealed narrowing of the retinal blood vessels, numerous pigmented spots in the middle periphery, and degenerated lesions in the macula (Figure 3C,D). ERG also disappeared in all stimuli (Appendix A). FAF imaging showed low fluorescence, which was consistent with retinal degeneration (Figure 3E,F). OCT showed that the retina was highly thinning, and the EZ line could not be observed at all (Figure 3G,H).

#### 3.2.2. Genetic Studies

Trio-based WES of family 2 revealed LOH stretches in the whole of chromosome 8, where eight rare homozygous candidate variants were found, and parental genotypes of these variants indicated maternal UPD (Figure 4A). Three of the eight genes are associated with human disorders; *RP1L1* is responsible for retinitis pigmentosa (RP) (MIM # 618826, Appendix A). However, the variant causes the in-frame deletion of poorly conserved amino acids, and many in-frame deletion/duplications have been reported in this region, suggesting that this *RP1L1* variant is less likely to be pathogenic. Alu insertion into exon 4 of *RP1*, which would be missed by WES, may be among the major causes of RP in Japan [27]. We carefully reviewed alignments with an integrative genomics viewer and found a sharp decline in the sequencing depth of exon 4 of *RP1* in patient 2 (Figure 4B, red arrow), suggesting the Alu insertion. This finding was also visible in the patient’s mother, although to a lesser extent, probably due to the low capture efficiency of the inserted fragments. De novo assembly of clipped reads suggested duplicated 11 bp segments, indicating a target site duplication, which is characteristic of retrotransposition events (Appendix A; [28]). PCR analysis confirmed homozygous insertion (c.4052_4053ins328, p.[Tyr1352Alafs]) in the patient and heterozygous insertion in his mother (Figure 4C).

Including these 2 cases, 11 cases of retinal dystrophy caused by UPD have been reported to date (Table 1). All previous reports, except our report, include the single-case spreading of chromosomes 1, 2, 6, and 14. Among them, four cases had UPD of chromosome 1. Various diseases, including RP (2), Stargardt’s disease (2), TULP1-related retinal dystrophy (2), and achromatopsia (2) have been reported. The eleven cases included three males, eight females, and six paternal and five maternal disomy cases. The cases of whole-chromosome UPD numbered more than partial UPD. Although most cases occurred in childhood, some adult cases have also been recognized. Refractive error was often observed and ranged from +4.00D to −8.00D. Two of the eleven cases were born with a low birth weight; nystagmus was observed in five of the eleven cases; and intellectual disability was observed in two of the eleven cases.

## 4. Discussion

UPD is a rare event in which an individual with a diploid genome carries either two homologs of a pair of chromosomes from one parent (uniparental heterodisomy) or two copies of a single chromosome from one parent (uniparental isodisomy) [1,4]. Recessive genetic diseases are rarely caused by UPD. There have been few reports of recessive retinal dystrophy caused by UPD on chromosomes 1, 2, 6, and 14 [9,10,11,12,13,14,15,16,17]. We performed WES in cases with early-onset retinal dystrophy and identified two cases of UPD on chromosomes 4 and 8. Both patients had maternal isodisomy. Based on literature review findings, this is the first report of retinal dystrophy caused by UPD on chromosomes 4 and 8. In case 1, a homozygous *SRD5A3* gene variant was detected, and *SRD5A3*-CDG was strongly suspected. We diagnosed the patient with CDG because the disorder of sugar chain addition to transferrin was confirmed. In case 2, although the Alu insertion of the *RP1* gene, which is common in Japanese *RP1*-related ARRP, could not be detected by WES, the homozygosity of the patient was confirmed by PCR amplification using previously reported primers.

Case 1 had intellectual disability and nystagmus from birth, and severe myopia and astigmatism were also observed. The patient was 14 years old, and her BCVA was 0.08. Both the rod and cone responses of ERG disappeared from early childhood. We considered color blindness as a diagnosis when the patient was 9 years of age because the visual field was relatively good. Although she had a history of epilepsy at the age of 2 years, it was well controlled, and her major problem was visual impairment.

CDG is a condition that causes systemic abnormalities; established in the last 30 years, it is accompanied by intellectual disability and neurological symptoms. Ophthalmic findings include strabismus, retinal degeneration, nystagmus, and myopia. In this study, WES revealed an abnormality in the *SRD5A3* gene, suggesting *SRD5A3*-CDG. MS employing transferrin, a typical glycoprotein, showed a deficiency in sugar chains, confirming the diagnosis of CDG. According to the Online Mendelian Inheritance in Man, clinical findings of *SRD5A3*-CDG include cerebral malformations, intellectual disability, coloboma, ichthyosis, and endocrine abnormalities. To date, few reports concerning the genotype and phenotype of the *SRD5A3* gene have been published. Ocular findings by CDG due to the *SRD5A3* gene variant have been reported by microphthalmia, cataract, chorioretinal coloboma, glaucoma, optic nerve hypoplasia/atrophy, and nystagmus [26,29]. The *SRD5A3* gene variant causes early-onset retinal dystrophy, in addition to optic nerve complications [30,31,32]. In our case, color vision deficiency at an early stage may be considered to involve optic nerve dysfunction as well as retinal dystrophy. Ophthalmological treatment for complications such as eye drops for glaucoma, surgery for cataracts, and school attendance support and low vision rehabilitation for better quality of life is recommended when the systemic symptoms are relatively mild.

*RP1* variants cause autosomal dominant RP (ADRP), autosomal recessive RP (ARRP), and autosomal recessive cone dystrophy/cone rod dystrophy (AR-COD/CORD). In the ADRP phenotype, all *RP1* variants are truncated within the hotspot region and express the truncated protein, suggesting a dominant-negative effect [32,33,34,35,36,37]. Siemiatkowska et al. [35] reported that the combination of *RP1* variants that cause *RP1*-related ARRP involves mainly truncated variants of two alleles outside the hotspot region, leading to the loss of *RP1* function. We examined the first Japanese *RP1*-ARRP case, in which an allele was difficult to judge inside or outside of the hotspot region [38]. Mizobuchi et al. [37] reported that the Alu element insertion of the *RP1* gene was the most frequent pathogenic variant (32.0%, 16/50 allele) in Japanese *RP1*-related ARRP and that Japanese *RP1*-related ARRP included homozygous Alu element insertion (5 of 26: 19%). Our case with homozygous Alu element insertion due to UPD is suspected to be the *RP1*-related ARRP phenotype. *RP1*-related ARRP exhibits early-onset, severe, and progressive visual impairment, with fundus findings including the narrowing of retinal blood vessels, pigmented spots, pallor of the optic nerve head, and flat ERG. In *RP1*-related ARRP, degeneration often involves the macula. Our case presented with childhood onset; the patient has used a white cane as a walking aid since the age of 20 years. Clinical findings, including fundus findings, extinguished ERG, and retinal degeneration involving macular lesions, were consistent with *RP-1*-related ARRP.

Although there have been reports of retinal dystrophy caused by UPD on chromosomes 1, 2, 6, and 14, it seems to concentrate on chromosome 1 (Table 1). It is necessary to accumulate UPD cases to clarify whether retinal dystrophy caused by UPD is most likely to occur on chromosome 1. In addition, although whole UPD is more frequently reported than partial UPD, previous reports are limited in terms of the number of haplotype markers. The current WES could determine the whole or partial UPD more precisely. Engle et al. [8] reported that UPD causes 40 recessive genetic diseases on 13 autosomal and XY chromosomes. To the best of our knowledge, this is the first report of retinal dystrophy caused by UPD in Japanese patients. Case 1 is the first case of Japanese *SRD5A3*-CDG. Our cases presented with features consistent with those of either CDG due to the *SRD5A3* gene variant or *RP1*-related ARRP. There have been limited reports of retinal dystrophy by UPD; thus, a literature review was warranted. Although retinal dystrophy by UPD varied from syndromic RP to achromatopsia, the clinical picture with UPD seems to be equivalent to that without UPD; therefore, UPD may be difficult to suspect as the cause based on the clinical picture alone.

## 5. Conclusions

We report two cases of retinal dystrophy caused by UPD. LOH was observed on chromosomes 4 and 8. Each case was diagnosed as *SRD5A3*-CDG and *RP1*-related RP, and the clinical picture was equivalent to that without UPD. There have been very few reports of retinal dystrophy caused by UPD, suggesting that it is rare. Genetic counseling may be encouraged in pediatric cases of retinal dystrophy to help manage various retinal dystrophies that develop in childhood.

## Figures and Tables

**Figure 1 genes-13-00359-f001:**
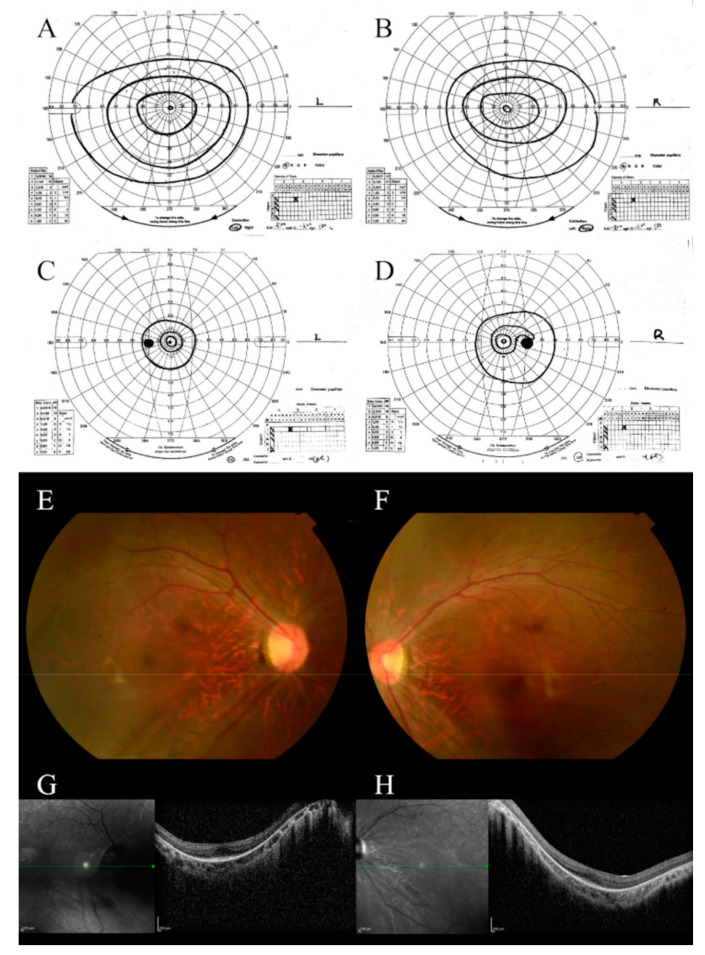
Clinical features of case 1. The visual field examined at the age of 9 years: (**A**) left eye; (**B**) right eye. Visual field examined at the age of 14 years: (**C**) left eye; (**D**) right eye. The visual field was progressively constricted. Fundus photograph obtained at the age of 9 years: (**E**) right eye; (**F**) left eye. Fundus photography showed poor retinal color and narrowing of the retinal blood vessels. OCT images obtained at the age of 10 years: (**G**) right eye; (**H**) left eye. Although OCT images were poor due to nystagmus, retinal thinning and EZ line disappearance were observed.

**Figure 2 genes-13-00359-f002:**
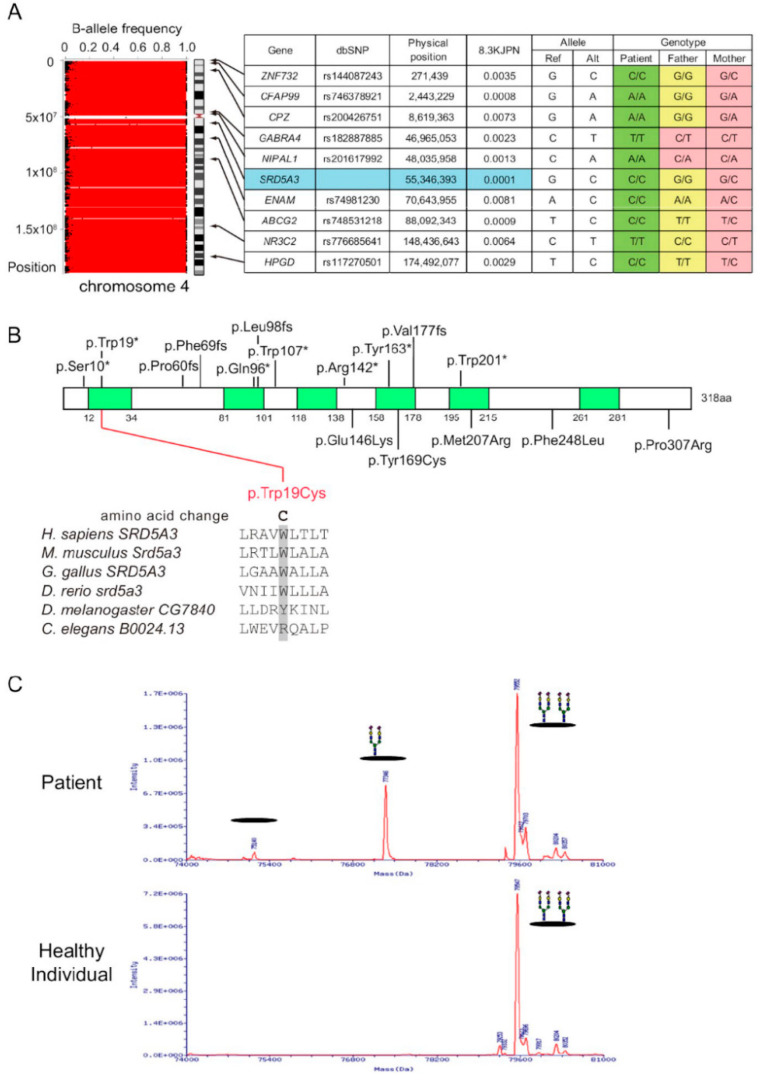
Findings of genetic and glycosylation analyses in patient 1. (**A**) (Left) H3M2 analysis using WES data showed LOH stretches on the whole of chromosome 4. (Right) Gene name, physical position, allele frequency in 8.3KJPN, and genotypes of the patient and her parents with 10 candidate variants are shown. Homozygous for reference allele, homozygous for alternate allele, and heterozygous genotypes are highlighted in yellow, green, and pink, respectively. Trio genotypes demonstrated maternal uniparental isodisomy. (**B**) Locations of reported *SRD5A3* variants. Human *SRD5A3* is 318 aa in length (NP_078868.1) and contains six putative transmembrane domains according to UniProt (Q9H8P0). Nonsense and frameshift variants are shown in the upper section, and missense variants are shown in the lower section. The p.T*rp1*9Cys variant was in the transmembrane domain and substituted a conserved amino acid among vertebrates. (**C**) Deconvoluted spectra of transferrin from a patient (upper) and healthy individual (lower), analyzed by ESI-MS. The presence of the molecules lacking one or two N-glycans indicates deficient N-glycosylation.

**Figure 3 genes-13-00359-f003:**
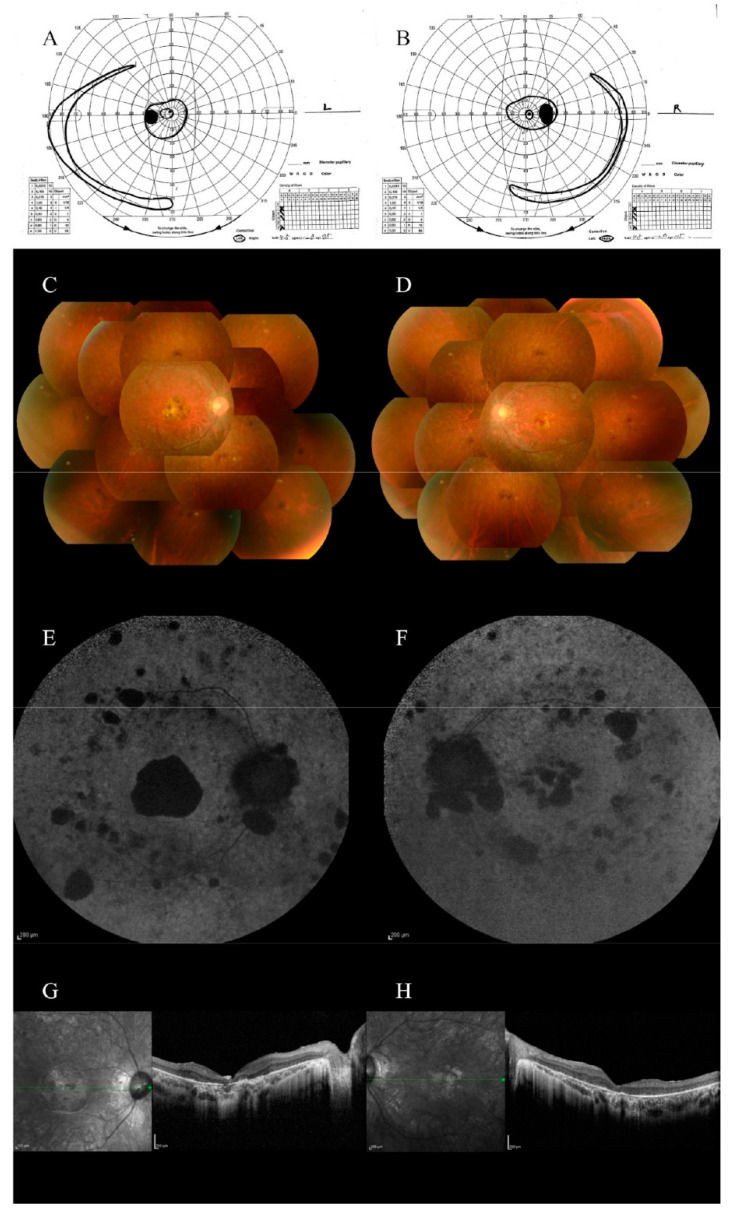
Clinical features of case 2. The visual field examined at the age of 26 years: (**A**) left eye; (**B**) right eye. The visual field was constricted. Fundus photograph obtained at the age of 26 years: (**C**) right eye; (**D**) left eye. Fundus photography showed poor retinal color and narrowing of the retinal blood vessels. FAF obtained at the age of 31 years: (**E**) right eye; (**F**) left eye. FAF showed low fluorescence, consistent with retinal degeneration. OCT images obtained at the age of 31 years: (**G**) right eye; (**H**) left eye. OCT images showed that the retina was highly thinning, and the EZ line could not be observed at all.

**Figure 4 genes-13-00359-f004:**
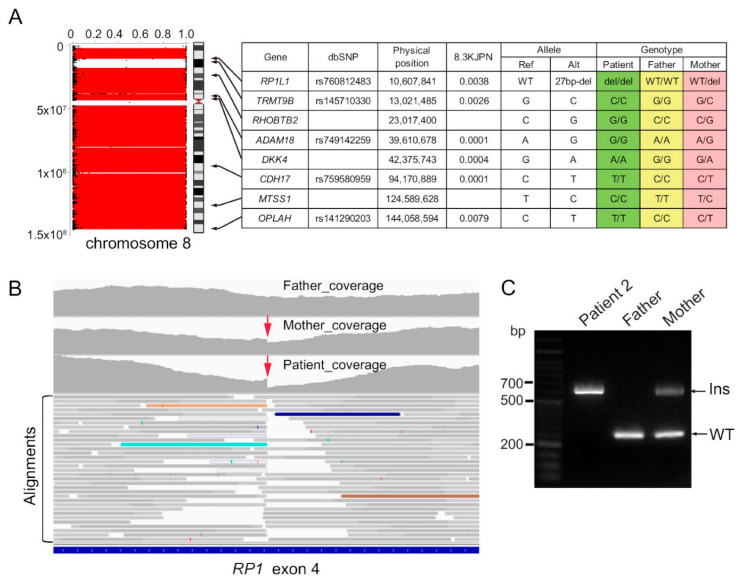
Genetic findings in patient 2. (**A**) (Left) H3M2 analysis using WES data showed LOH stretches on the whole of chromosome 8. (Right) Gene name, physical position, allele frequency in 8.3KJPN, and genotypes of trio samples of eight candidate variants are shown. Homozygous for reference allele, homozygous for alternate allele, and heterozygous genotypes are highlighted in yellow, green, and pink, respectively. Trio genotypes indicated maternal uniparental isodisomy. (**B**) Integrative genomics viewer showed a sharp decline in sequencing depth at exon 4 of *RP1* (red arrows), suggesting the Alu insertion. Discordant reads are highlighted in colors. (**C**) PCR analysis confirmed homozygous Alu insertion in the patient, and his mother was a carrier of the insertion.

**Table 1 genes-13-00359-t001:** Previous reports of uniparental disomy (UPD) in retinal dystrophy.

CHR	Sex	Onset	Disomy	Gene	Origin	Ocular Diagnosis	Nystagmus	Ocular Findings	Intellectual Disability	Other Systemic Abnormalities	References
1	male	<5 years	Whole	*RPE65*	paternal	LCA	+	VA: 20/200 in both eyes with a refraction of +4.00 (8 y) ERG: rod and cone responses were not detectable Prognosis: small islands of vision remaining (50 y)	-	hyperbetalipoproteinemiabenign hyperbilirubinemia	[10]
1	female	-	Partial	*USH2A*	paternal	RP	-	VA: 20/40 in both eyes with a mean refraction of +1.50 (49 y) VF: V4e were constricted to a central island extending to the 8° ERG: rod-plus-cone and cone ERG amplitudes were remarkably decreased	-	no hearing loss	[11]
1	female	15 years	Whole	*ABCA4*	paternal	Stargardt disease	-	VA: 20/200 OD and 20/150 OS Fundus: macular and extramacular pisciform yellow flecks VF: bilateral central scotomas ERG: normal	-	-	[12]
1	female	2–3 years	Partial	*ABCA4*	paternal	Stargardt disease	-	mild vision loss and strabismus (2–3 y), photophobia and dyschromatopsia VA: 100/200 Fundus: yellowish flecks at the macula	-	-	[14]
2	female	3–5 years	Whole	*MERTK*	paternal	RP	-	night blindness, poor vision (preschool), peripheral vision reduction Prognosis: 5° visual field (34 y)	-	-	[10]
2	female	first week	Partial or Whole	*CNGA3*	paternal	ACHM	+	photophobia Color vision tests: findings typical for ACHM ERG: severely reduced cone and responses Prognosis: Visual disturbances remained stable	-	multiple bilateral kidney cysts with a maximum diameter of 14 mm	[17]
4	female	birth	Whole	*SRD5A3*	maternal	RD	+	Fundus: poor retinal color and narrowing of the retinal blood vessels VF: remarkably constricted (14 y) ERG: extinguished pattern	+	epilepsy	this study
6	male	43 years	Whole	*TULP1*	maternal	RD with cone dysfunction	-	severe color vision defects VA: 80/200 OD and 120/200 OS (at onset) VF: central scotoma Fundus: macular bull’s eye, peripheral mottling vessels ERG: relatively preserved ERG Prognosis: CF OD and 10/200 OS (52 y)	-	intrauterine growth retardation	[15]
6	female	3 months	Whole	*TULP1*	maternal	RD with rod-cone dysfunction	+	poor pupillary constriction to strong light retinal pallor (3 months) poor night vision and peripheral vision VA: 6/76 in both eyes Fundus: RP-like, no optic disc atrophy ERG: non-recordable (17 months)	-	intrauterine growth retardation	[16]
8	male	7 years	Whole	*RP1*	maternal	RP	-	photophobia and night blindness (7 y) VF: highly constricted Fundus: typical findings RP, and degenerated lesions in the macula FAF: low fluorescence consistent with retinal degeneration ERG: extinguished pattern	-	-	this study
14	female	5 days	Partial	*CNGB3*	maternal	ACHM	+	A-pattern exotropia, sluggish pupils without afferent defect progressive compound myopic astigmatic refractive error VA: 20/160 in both eyes Color vision test: findings typical for ACHM ERG: no recordable cone function, normal rod function	+	short stature, minimal dysmorphism premature puberty, small hands and feet reproductive history of three consecutive first-trimester miscarriages	[9,13]

CHR: chromosome, LCA: Leber congenital amaurosis, VA: visual acuity, ERG: electroretinography, RP: retinitis pigmentosa, VF: visual field, ACHM: achromatopsia, RD: retinal dystrophy.

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
