# Peer review of "Maternal Uniparental Isodisomy of Chromosome 4 and 8 in Patients with Retinal Dystrophy: SRD5A3-Congenital Disorders of Glycosylation and RP1-Related Retinitis Pigmentosa"

_genes, 2022, doi:10.3390/genes13020359_

Round 1
Reviewer 1 Report
The authors report on a very rare genetic cause for retinal dystrophy. They report two cases of diagnosed retinal dystrophy and found the cause being uniparental disomy (UPD).
Additionally the authors performed a literature search reporting a summary of the reports about retinal dystrophy caused by UPD. I would like to compliment the authors with this interesting work adding yet another genetic cause for retinal dystrophy solving this puzzle.
Below you find some comments and suggestions.
- The outline of the abstract could be improved. I suggest the authors follow the flow: purpose, methods, results, conclusions. In other words: add some structure.
- “We performed whole exome sequencing in cases…..”. What were the criteria of these cases? How many in total? Were these cases not diagnosed? Causative mutation not known? Screens with known causative mutations were performed? Etc.
- Methods: 2.3: up to now it is not clear why the authors performed this analysis. Maybe add it to the intro? Or explain why it was performed in M&M.
- Methods: add the systematic search term that was used to get the papers from PubMed. Also add inclusion and exclusion criteria. If the authors performed a systematic search a protocol should have been written as well before performing the search. How to handle the results etc. Also looking at bias. If this is not the case, maybe not add the word “systematic”.
- Results section: maybe describe the results per case as a whole story. Now it reads a little weird. For example, 3.2 comes after the description of case 2. Although the analysis belongs to case 1. I would suggest to divide the results in Case 1 and Case 2.
- Fig 2: “Colors for genotypes are same as Figure 1”… I would suggest that the authors add this description again to the figure.
- Especially for Case 2 (diagnosed with RP): were no previous genetic tests performed? If there were, I would suggest that the authors report on the genes that were checked.
- Conclusions: The authors repeat many statements from the results section. I would suggest that (per case) an in depth analysis and comparison is made to other known cases. Avoid repeating sentences and information.
- “Although there have been reports of retinal ……….. it seems to concentrate on chromosome 1”. Can you really conclude this given the very limited number of cases described?
- Conclusion: Correct the sentence: Each case was diagnosed as ……. RP.
- Conclusion: help manage various retinal dystrophies: What would the authors suggest as options to manage the retinal dystrophies that they have reported in Table 1 ?
- Would the authors suggest to add UPD analysis to the general panel of genetic testing for retinal dystrophies? Or do they suggest a certain sequence of genetic tests being done?
Reviewer 2 Report
The authors describe two cases of maternal uniparental isodisomy in patients with Retinitis Pigmentosa. Whole exome sequencing (WES) was performed to identify genetic changes. In one of the cases, WES was non-confirmatory so Alu insertion was confirmed using PCR. In addition to sharing clinical phenotype in patients, the authors also discuss other cases of parental disomy known through previously published work. Overall, it is a well-described work and it is important for the field to know about rare genetic cases. Table 1 is very useful. Following are the comments on the manuscript:
- The uploaded file does not have Supplemental figures and tables referred to in the text. Only legends are included in the text.
- Was sequencing performed on the PCR amplified fragment? The sequence would confirm that the increase in size is due to Alu insertion. It seems that this information was supposed to be in the Supplementary section, which is missing in the manuscript.
- Usually the clinical characteristics are described first followed by further analyses, so in that regard it is interesting to see that clinical characteristics are described last in this manuscript.
- Figure 1C is explained in Results section after Figure 2. Under ‘Genetic studies’ section the authors write “SRD5A3 encodes steroid 5α-reductase 3, which is essential for N-glycosylation by converting polyprenol to dolichol” but it would be good to add a sentence or two under ‘Mass spectrum of transferrin’ explaining why transferrin glycosylation was tested.
- Figure 1 and 2 are needed at high resolution. Both are very difficult to read.
- Line 58: WES is Whole Exome-sequencing. Please replace exosome with exome.
- Line 197: “with almost the same power from the age of 8 to 14 years”. ‘Acuity’ is a preferred word instead of ‘power’
- Lines 277-278: “Ophthalmology treatment is recommended when the systemic symptoms are relatively mild.” Please provide examples of what kind of ophthalmology treatments could be useful.
- Figure description in text would be better as 3A-D instead of 3ABCD.
